# Optimization of Novel Naproxen-Loaded Chitosan/Carrageenan Nanocarrier-Based Gel for Topical Delivery: Ex Vivo, Histopathological, and In Vivo Evaluation

**DOI:** 10.3390/ph14060557

**Published:** 2021-06-11

**Authors:** Sobia Noreen, Fahad Pervaiz, Akram Ashames, Manal Buabeid, Khairi Fahelelbom, Hina Shoukat, Irsah Maqbool, Ghulam Murtaza

**Affiliations:** 1Department of Pharmaceutics, Faculty of Pharmacy, The Islamia University of Bahawalpur, Bahawalpur 63100, Pakistan; sobianoreen07@gmail.com (S.N.); hinashoukat50@yahoo.com (H.S.); irsamaqbool@yahoo.com (I.M.); 2Department of Pharmaceutical Sciences, College of Pharmacy and Health Sciences, Ajman University, Ajman P.O. Box 346, United Arab Emirates; a.ashames@ajman.ac.ae; 3Medical and Bio-Allied Health Sciences Research Centre, Ajman University, Ajman P.O. Box 346, United Arab Emirates; 4Department of Clinical Sciences, College of Pharmacy and Health Sciences, Ajman University, Ajman P.O. Box 346, United Arab Emirates; 5Department of Pharmaceutical Sciences, College of Pharmacy, Al Ain University, Al Ain P.O. Box 64141, United Arab Emirates; khairi.mustafa@aau.ac.ae; 6Department of Pharmacy, COMSATS University Islamabad, Lahore Campus, Lahore 54000, Pakistan

**Keywords:** chitosan, carrageenan, anti-inflammatory, naproxen, polyelectrolyte complexation, nanocarriers, Ca-940 gel, toxicity study

## Abstract

Naproxen (NAP) is commonly used for pain, inflammation, and stiffness associated with arthritis. However, systemic administration is linked with several gastrointestinal tract (GIT) side effects. The present work aims to prepare and evaluate NAP nanoparticulate shells of chitosan (CS) and carrageenan (CRG) loaded into a Carbopol 940 (Ca-940) gel system with unique features of sustained drug delivery as well as improved permeation through a topical route. Moreover, this study aims to evaluate its ex vivo, histopathological, and in vivo anti-inflammatory activity in albino Wistar rats. The percentage of ex vivo drug permeation patterns in the optimized formulation (No) was higher (88.66%) than the control gel (36.195%). Oral toxicity studies of developed nanoparticles in albino rabbits showed that the NAP-loaded CS/CRG are non-toxic and, upon histopathological evaluation, no sign of incompatibility was observed compared to the control group. A In Vivo study showed that the optimized gel formulation (No) was more effective than the control gel (Nc) in treating arthritis-associated inflammation. The sustained permeation and the absence of skin irritation make this novel NAP nanoparticle-loaded gel based on CS/CRG a suitable drug delivery system for topical application and has the potential for improved patient compliance and reduced GIT-related side effects in arthritis.

## 1. Introduction

Rheumatoid arthritis (RA) is an inflammatory disease associated with severe pain, stiffness, and peripheral joint swelling. Inflammatory events are initiated from the interaction of antigen-presenting cells (APCs) with CD+T cells. Complex cell–cell interaction leads to macrophage activation with an enormous release of proinflammatory cytokines such as IL-1 and TNFα. These cytokines activate synovial fibroblasts and chondrocytes in surrounding articular cartilage and release enzymes that destroy proteoglycans and collagen, causing tissue destruction [1]. RA mainly occurs in women as compared to men. Around 1–2% of the general population is affected worldwide by arthritis [2]. Non-steroidal anti-inflammatory drugs (NSAIDs) are commonly prescribed to treat inflammatory, acute, and chronic pain conditions [3].

Naproxen (NAP), (2S)-2-(6-methoxynaphthalen-2-yl) propanoate, belongs to NSAIDs’ propionic acid class. It is generally used to treat pain, pyrexia, inflammation, and stiffness produced by osteoarthritis, rheumatoid arthritis, injuries, tendinitis, bursitis, and psoriatic arthritis [4]. Therefore, the designing of naproxen’s formulation with an improved controlled release pattern will possibly have exceptional advantages in treating the body’s inflammatory and painful states [5].

Anti-inflammatory effects of NAP are mediated by the inhibition of COX-1 and COX-2, which are responsible for prostaglandin E2 production when activated by inflammatory mediators such as tumor necrosis factor and interleukins [6]. The use of oral NSAIDs can damage the gastrointestinal tract, leading to peptic ulcers and hemorrhagic disorders [7]. Analogous to other NSAIDs, naproxen also causes gastric bleeding and ulceration after oral administration. The mechanisms underlying these gastric damage events include prostaglandin-mediated, increased gastric acid secretion, reduced mucus, bicarbonate secretion, and decreased mucosal cell proliferation and blood flow [8]. 

From the perspective of the side effects related to naproxen’s oral route, it can be administered safely via topical drug delivery with minimal side effects, including peptic ulcer disease and GI hemorrhage. Additionally, the topical approach offers certain distinct advantages: a practically larger surface area of skin for absorption, local drug delivery to affected tissues, a non-invasive route, eliminated side effects, maintenance of plasma–drug concentration, ease of removal or replacement, and the avoidance of presystemic metabolism [9]. Typically, a polyelectrolyte complex (PEC) is the formation of the complex in a reaction of a polyanion (PA−) with counter cations (CC+), and a polycation (PC+) with counter anions (CA−) [10]. CS is a naturally occurring polysaccharide with [(1–4) 2-amino 2-deoxy-β-d-glucan] linkage [11]. Owing to its outstanding characteristics, including nontoxicity, biodegradability, biocompatibility [12], and gel- and film-forming properties [13], it has been broadly employed in the fabrication of novel polymeric drug delivery systems [14]. Its anti-inflammatory, antioxidant, and anti-microbial characteristics rank CS as an ideal vehicle for transdermal drug delivery [15]. Various studies reported the behavior of CS for TDD applications [16,17,18,19]. Bhaskar et al. prepared solid lipid nanoparticles (SLNs) and nanostructured lipid carriers (NLCs) for enhanced transdermal delivery of flurbiprofen by using chitosan as a carrier [20]. CRG is a linear anionic polysaccharide that contains many sulfate groups in the galactose dimers [21].

In the biomedical and pharmaceutical sciences, the blending of nanoparticles (NPs) with CRG magnifies their properties, and the sustained release characteristics are improved due to carrageenan’s gelling properties [22]. Negatively charged carrageenan reacts with positively charged chitosan in the crosslinker’s presence to form CS/CRG sustained release nanoparticles. The positive surface charge and size of CS/CRG nanoparticles are suitable for penetrating epithelial surfaces [23]. Therefore, CRG has been used in several studies to develop a transdermal delivery system [24].

The current study aimed to develop a novel naproxen-loaded CS/CRG polymeric nanoparticulate-based gel system using a polyelectrolyte complexation technique. Naproxen-loaded chitosan/carrageenan nanoparticles were incorporated into Ca-940 gel and evaluated for transdermal delivery of naproxen by ex vivo permeation and in vivo anti-inflammatory studies. The histopathological slides also performed an acute oral toxicity study. To the best of our knowledge, no study has incorporated NAP-loaded CS/CRG nanoparticles in the gel for TDD application in inflammatory conditions such as arthritis to minimize systemic side effects of NAP.

## 2. Results and Discussion

### 2.1. Preparation and Optimization of Naproxen-Loaded Nanoparticles

NAP-loaded CS/CRG NPs were successfully prepared by the polyelectrolyte complexation method. Once the polymer-containing solutions were mixed, PEC formation occurred among the amino groups of CS, having a positive charge. The sulfate and phosphate groups of CRG and TPP, respectively, having a negative charge, accelerated nanoparticle formation. STPP promotes a strong interaction, as it offers a cross-linking effect. At a higher percentage (>0.1%) of CRG, precipitation was observed, and a further increase in CRG (1%) led to clump formation. However, when the lower percentage of CRG was employed, no nanoparticle formation occurred. The high concentration of CRG resulted in less engagement of amino groups in neutralization with the sulfate group, causing precipitation. On the contrary, a lower concentration of carrageenan did not support nanoparticle development due to the insufficient volume of counter ions. After washing away the unreacted components and freeze-drying the concentrated nanoparticle suspension, dried nanoparticles were obtained [25].

### 2.2. Nanoparticles Characterization 

#### 2.2.1. Scanning Electron Microscopic Analysis

SEM analysis for the NAP-loaded CS/CRG nanoparticle formulations (lyophilized and in water dispersion) was performed. Figure 1a,c demonstrates the SEM microphotographs of N11 (in water dispersion) and the optimized formulation (No) (in water dispersion), respectively. While Figure 1b,d depicts the SEM images of N11 (lyophilized) and the optimized formulation (No) (lyophilized), which confirmed a compact and solid structure, presenting the tendency to display a spherical shape [26].

#### 2.2.2. FTIR Spectra of Naproxen-Loaded Nanoparticles Formulation

FTIR spectra of NAP showed absorption bands at 1028, 854, 1630, and 1727 cm^−1^. The vibration modes were identified at 1028.96 and 854.78 cm^−1^ parallel to C–O–C bonds in NAP. Moreover, the band at 1733.52 cm^−1^ was parallel to the carbonyl stretching region. NAP absorption bands at 1028, 854, 1630, and 1727 cm^−1^ in the carbonyl stretching region remained unchanged, specifying that there was no drug–polymer hydrogen bonding interaction [27,28,29]. The FTIR analysis of CS revealed an amide I peak at a range of 1700–1500 cm^−1^ (1652.66) and an amide II peak at 1586.71 cm^−1^. The broadband at 3366.08 cm^−1^ corresponded to the stretching vibration of hydroxyl groups at a range of 3400–3300 cm^−1^. The broad peak at 1162.57 cm^−1^ represented asymmetric stretching of C–O–C in a glycosidic linkage, and the peak at 1029.32 cm^−1^ presented the stretching vibration of C–O [30]. FTIR spectra of carrageenan at 925.16 cm^−1^ were attributed to the C–O–C vibration of the 3, 6-anhydro-d-galactose residue. The intense band at 1647.04 cm^−1^ linked with the structural water deformation band [31].

A new absorption band confirmed the development of the CS/CRG polyelectrolyte complex at 1505 cm^−1^ due to the appearance of the -NH3 group. Moreover, both amide peaks of CS transformed into a singlet band at 1630.90 cm^−1^, and NH2 groups were also recognized (Figure 2). Characteristic peaks of CRG were recognized in the NPS spectrum, such as sulfate groups at 1263.93 cm^−1^, 3, 6-anhydrogalactose at 925.31 cm^−1^, and galactose-4-sulfate at 854.56 cm^−1^. This indicated the interaction between protonated amine groups of CS and sulfate groups of carrageenan in the successful formation of the CS/CRG polyelectrolyte complex. No prominent change was observed in the peaks of drug, polymer, and physical mixtures, specifying the absence of any hydrogen bonding interaction between drug and polymer. The NAP-loaded CS/CRG nanoparticle optimized formulation spectra showed no interaction between protonated amine groups of CS and sulfates groups of carrageenan in the successful formation of the CS/CRG polyelectrolyte complex [27,32]. 

#### 2.2.3. Powdered X-ray Diffraction (pXRD)

In Figure 3, the pXRD diffraction patterns of NAP, CS, CRG, and drug-loaded CS/CRGs nanoparticles are displayed. The diffraction pattern of pure NAP showed the crystalline nature of the drug and distinct diffraction peaks at 2θ value of 15.2°, 17.0°, 18.08°, 20.70°, 24.40°, and 28.3°, which are characteristic peaks for NAP [33]. The diffraction pattern of CS shows its semi-crystalline nature and revealed a reflection fall at 10° and 20° [34]. The XRD diffraction pattern of pure carrageenan exhibits sharp peaks at 2θ = 20°, 23.05°, 22.35°, 28.79°, 31.70°, 32.82°, 39.81°, and 45.66°. These peaks represent the amorphous nature of the polymer [35]. Crystallinity peaks of pure drug and polymers are observed in the XRD analysis of physical mixture (PM) representing no interaction among the formulation components. In the drug-loaded polyelectrolyte complex nanoparticles, a diffuse pattern with no precise, sharp peaks is observed. The PEC presents a reduction in NAP peak intensity, showing the existence of an amorphous form when loaded with CS/CRG nanoparticles.

#### 2.2.4. Entrapment Efficiency

Formulation N11 showed the lowest EE (93.33%), while N1 exhibited maximum entrapment efficiency of 97.55% (Table 1). It was revealed by ANOVA (Table 2) that the EE exhibited significant differences when changing CRG concentration (X1) and drug concentration (X2), while a non-significant effect was observed when altering stirring speed (X3). 

The quadratic expression relating the EE with independent variables is presented in Equation (1):Entrapment Efficiency (Y1) = + 95.20 + 1.69 × A − 0.58 × B − 0.12 × C + 0.025 × A × B − 0.065 × A × C − 0.097 × B × C + 0.26 × A2 + 9.750E − 003 × B2 − 0.090 × C2 (1)

A significant (*p* < 0.05) increase in the EE was noticed with the increased concentration of CRG (Table 2 and Figure 4) in the preparation of drug-loaded CS/CRG nanoparticles. This is due to the increase in CRG concentration increased the internal phase’s viscosity and thickness, leading to ionic cross-linking of CRG with free CS in the dispersion. The formulations showed a significant (*p* < 0.05) increase in drug entrapment by decreasing the formulations’ drug amounts. This is because a high amount of drug increases the solvent’s viscosity, resulting in larger-sized particles and increasing the hardening time of the particles, which provided sufficient time for the drug to move out of particles. Figure 4 reveals a non-significant decrease (*p* > 0.05) in the EE by increasing stirring speed from 600 to 900 rpm. This is because smaller particles have a large surface area, and there was further diffusion of the drug into the continuous phase [36,37,38]. 

#### 2.2.5. Zetasizer, Polydispersity Index, and Zeta Potential Determination of NAP-Loaded Nanoparticles

NAP-loaded CS/CRG nanoparticle size depends on three variables: carrageenan concentration, drug concentration, and complexation time. Formulation N15 displayed maximum particle size (378.0 ± 84.62 nm) (Figure 1), whereas minimum particle size was observed in formulation N10 (156.78 ± 45.53 nm) (Table 1). The polydispersity (PDI) value of the CS/CRG nanoparticles ranged between 0.308–0.521, thus representing a narrow and promising particle size distribution (PDI < 0.5) (Table 1). The PDI usually increased according to the increase in the CS to CRG mass ratio [39,40]. CS/CRG nanoparticles showed positive zeta potential, having a range between +25 and +35 mV (Table 1).

The impact of various independent variables, including polymer concentration, drug concentration, and stirring speed, was significant, as determined by ANOVA. A quadratic Equation (2) relating the particle size with independent parameters was employed as follows.
Particle Size = + 264.33 + 91.33 × A + 14.81 × B + 10.80 × C + 5.40 × A × B − 0.35 × A × C + 0.095 × B × C − 2.98 × A2 + 3.62 × B2 − 1.39 × C2(2)

Results revealed that formulations with 0.05% CRG gave smaller size particles than formulations containing 0.06% and 0.07%. Formulations N4, N10, N11, and N17 exhibited smaller particle sizes, 162.75, 156.78, 176.68, and 188.32 nm, respectively, due to a lower polymer concentration used, providing fewer binding sites for the crosslinker. Formulations N1, N3, N7, and N15 exhibited large particle sizes, 342.45 nm, 332.29 nm, 362.43 nm, and 378.0 nm, respectively, due to an increase in the CRG concentration. A substantial increase in CS/CRG percentage is the indicator of larger particle sizes. This is quite reasonable, as CRG is a bulky polymer and, therefore, larger quantities (0.07%) are directed to the enlargement of particles [41]. As the percentage of the loaded drug changed from 30% to 50%, the particle size increased significantly (*p* < 0.05) (Table 2 and Figure 5); accordingly, a high amount of drug-loaded nanoparticles corresponded with large particle size. The reason is that a high amount of drug during the complex formation caused the formation of the gap between CS, CRG, and TPP and decreased CS/CRG/TPP interaction; hence, large-sized particles were produced [42]. The effect of stirring speed was determined by varying the speed between 600 and 900 rpm, and it was found to be significant (*p* < 0.05). Smaller particle size was observed at slow stirring, while under vigorous shaking, larger-sized particles were formed. Size reduction is credited to the cross-linking effect; gentle stirring increased the condensation of polymeric chains, and successively smaller particles were produced. The stirring speed and particle size (Y1) were directly related (Figure 5). Owing to high shear energy and rapidity, the viscosity of the solution increased, which contributed to the aggregation of the smaller particles into larger-sized particles [43,44]. Some deviations in size distribution were observed due to the influence of oppositely charged polyions in the complex formation process throughout the formation of NPs [45].

### 2.3. Ex Vivo Permeation Study

Fabricated CS/CRG nanoparticles displayed a sustained permeation of NAP for 48 h. Ex vivo permeation pattern was biphasic; initially, fast permeation of NAP was noticed in the first 2 h, and afterward, slow and sustained drug permeation for 48 h was observed. Percentage cumulative drug permeation of all formulations (N1–N19) is provided in Table 1. The initial fast permeation from nanoparticles was conceivably owing to drug adsorption on the nanoparticles’ surfaces; however, as time proceeds, NAP may persistently permeate the polymer matrix (Figure 6). Consequently, the polymer matrix’s erosion to hydration showed higher permeability and flux than the NAP control gel [37]. 

A quadratic equation relating permeation with independent variables was:Permeation = + 89.66 − 2.70 × A − 0.55 × B − 1.51 × C − 0.13 × A × B − 0.45 × A × C + 0.12 × B × C + 0.13 × A2 − 0.26 × B2 + 0.32 × C2(3)

Ex vivo studies revealed that the permeation of drug nanoparticles into the skin was within the range of 85.47–93.66%. Figure 7 and Table 2 reveal significantly (*p* < 0.05) slower drug permeation for cross-linked NPs prepared at higher CRG concentration due to a higher degree of cross-linking. Furthermore, increasing the concentration of CRG (kappa) led to a larger particle size with a small surface area. Drug permeation from the nanoparticles was typically reliant on the loaded drug amount. Drug permeation was higher in the NPs of lesser drug entrapment efficiency. However, drug concentration had an insignificant effect (*p* > 0.05) on the nanoparticles’ drug permeation. Results specify that drug permeation was independent of the quantity of drug entrapped in the formulation. Drug permeation was slower and significant (*p* < 0.05) for nanoparticles prepared by increasing the stirring speed from 600 to 900 rpm because small particle size has a large surface area for the permeation of the drug [37,46,47].

The cumulative drug permeation percentage from all formulations in descending order is: 

N15 > N7 > N3 > N1 > N6 > No> N9 > N14 > N5 > N2 > N8 > N12 > N13 > N16 > N11 > N17 > N10 > N4 > N18 > N19.

### 2.4. Characterization of Carbopol 940 Gel 

#### 2.4.1. Appearance, Spreadability, pH, Viscosity, and Drug Content

The NAP-loaded nanoparticle gel formulations were examined visually for their color and spreadability. All NAP-loaded, NP-containing gels were transparent with a clear appearance and a smooth, homogenous texture. All gel formulations were easily spreadable with low shear force. All the formulation results shown in Table 3 indicate that all the polymers produced good gel spreadability by a small amount of shear force [48]. All NAP-loaded, nanoparticle-containing gel formulations had a pH in the acceptable range of 6.2–6.8 to avoid skin irritation [49]. Rheological studies of the NAP-loaded, nanoparticle-containing gel were performed, and results revealed that the viscosity of all formulations was in the range of 5102–5458 cps [37,50]. The drug content in all the gel preparations was in the range of 85% to 96%, which is an acceptable range for optimized therapeutic activity [51]. Results of all these evaluation parameters are shown in Table 3.

#### 2.4.2. Skin Irritation Studies

Skin irritation studies were completed to find the dermal toxicity of all developed formulations and control gel. All the gel preparations showed a Draize score of up to 1, i.e., slight erythema (light pink), indicating the tolerability and lower irritation potential for topical delivery [52] (Table 3).

#### 2.4.3. Stability Studies for NAP-Loaded Gel 

According to ICH norms, the study of the formulated gel preparations’ accelerated stability was performed at different temperature conditions. Stability data of formulations at (4 ± 1 °C) exhibited good stability behavior regarding pH, viscosity, appearance, spreadability, and percentage of drug content. Formulations at different temperatures (room temperature and accelerated temperature) became unstable [53]. 

### 2.5. Kinetics of Drug Permeation 

The value of the correlation coefficient was observed for zero-order, first-order, and Higuchi models, and the value of “*n*” exponent in the Korsmeyer–Peppas model was also applied (Table 4). A comparison of calculated values indicated that the zero-order model was the best fit compared to the first-order model in all formulations. A molecular diffusion release pattern was observed after fitting the data to the Higuchi model. In the Korsmeyer–Peppas model, the diffusional exponent “*n*” demonstrated the anomalous (non-Fickian) mechanism of drug permeation from the spherical nanoparticulate matrix [54,55]. 

The results of the drug permeation through full-thickness rat skin confirmed that NAP was released from the formulation, permeated through the FT rat skin, and could permeate through human skin [56].

### 2.6. Toxicity Study in Rabbits

Histopathological, biochemical, and hematological studies were performed for the determination of toxicity in the rabbit model. After 14 days of acute oral toxicity studies, the results of various parameters of biochemical, hematological, and weight variation studies in Group I (control) and Group II (treatment) are reported in Table 5. The results of the biochemical and hematological analyses of naproxen-loaded, CS/CRG-nanocarrier-based gel (Group II) presented insignificant changes in comparison to the control group. Histopathological evaluation within organs showed the non-toxic effect of CS/CRG nanocarriers (Figure 8). Seven vital organs, namely, the heart, intestine, kidney, liver, lungs, spleen, and liver, were evaluated, which show the absence of lesions, distortion, and no signs of toxicity at the cellular level (Table 5). Briefly, the acute oral toxicity study predicted that fabricated CS/CRG nanocarriers are useful as effective drug delivery vehicles due to their wide applications through various routes.

### 2.7. Optimization

#### 2.7.1. Evaluation of Optimized Formulation of Naproxen-Loaded CS/CRG Nanoparticles 

An optimized formulation was developed by applying the independent variables suggested by Design-Expert software (polymer 0.07%, drug 30%, and stirring speed 900 rpm). The values of entrapment efficiency, particle size, and drug permeation (%), as predicted by Design-Expert, were 97.43%, 345.015 nm, and 85.735%, respectively. Optimized formulation revealed an entrapment efficiency of 95.26% ± 3.23, the particle size of 355.7 ± 79.8 nm, and the zeta potential of +25 ± 3.1 mV, while 88.66% drug permeated from the nanoparticle-containing gel following a 48-h period. Therefore, the optimized nanoparticle-containing gel preparation is anticipated to sustain NAP permeation for 48 h (Table 6).

#### 2.7.2. Evaluation of Optimized Formulation of Naproxen-Loaded, CS/CRG-Nanoparticle-Containing Carbopol 940 (Ca-940) Gel

The optimized formulation of NAP-loaded nanoparticles was incorporated into 0.1% Carbopol 940 gel and used for further characterization. Stability studies were performed to check the interaction between the drug and Carbopol gel, which was necessary for obtaining an optimized formulation with an optimized therapeutic effect (Table 6). 

The ex vivo drug permeation of the optimized nanoparticle-containing gel preparation was compared with the control gel. Ex vivo drug discharge of NAP from the optimized nanoparticle-containing gel formulation presented improved drug permeation (88.66%) through the epidermis compared with the control gel (36.19%), as depicted in Figure 6, as the chitosan polymer in an NP formulation enhances permeation [57]. For the absolute permeation of nanoparticles through human skin, the nanoparticle size should be less than 400 nm for drug delivery applications [58]. Hence, the synthesized CS/CRG NPs are suitable for loading on and permeating NAP into the human skin. 

The optimized nanoparticle-containing gel preparation is anticipated to sustain NAP permeation for 48 h (Table 6). Ex vivo drug permeation data of the optimized formulation (No) were exposed to kinetic modeling, and they demonstrated that the permeation of NAP from optimized nanoparticles follow a non-Fickian mechanism, as confirmed by the value of the diffusional exponent (*n* = 0.873), and followed zero-order kinetics (R^2^ = 0.9927) better than first-order (R^2^ = 0.9818).

### 2.8. In Vivo Anti-Inflammatory Studies on Rats

Paw volume increased significantly (*p* < 0.05) in the formalin-induced edema (FIE) model. However, treatment with NAP NP gel reduced the inflammation and reduced paw volume significantly (*p* < 0.05) compared to the NAP control gel (Figure 9). NAP NP gel showed significant inhibition of 36%, 57%, and 79% at 1, 3, and 5 h, respectively. Reduction in paw volume also appeared in the NAP control gel group (21%, 45%, and 59%, respectively) (Table 7). In conclusion, the NAP NP gel has more significant anti-inflammatory effects due to enhanced permeation compared to the NAP control gel to treat arthritis. The same results in terms of percentage of inhibition of paw edema were achieved by Cong et al. when they compared the NAP and Indomethacin effects on the induction, duration, and intensity of rat paw edema [59]. Histopathological examination of rat paw tissue (Figure 10) showed increased inflammatory cells, severe edema, loosening of the epithelial layer, and accumulation of collagenous materials. Rats treated with NAP nanoparticle (NP) gel showed mild edema, and most of the inflammatory histological changes were set to normal at the end of the fifth hour (Figure 10D). However, the group treated with the NAP control gel showed marked histological changes (Figure 10C) in the accumulation of collagenous tissues in the deep dermis and infiltration of inflammatory cells compared to the group treated with the NAP NP gel. The NAP NP gel’s increased anti-inflammatory effects might be due to chitosan/carrageenan’s permeation enhancement effects [60].

## 3. Materials and Methods

### 3.1. Materials

Naproxen (PubChem CID: 23681059) was received as a kind gift from Schazoo Laboratories (Pvt.) Ltd. (Karachi, Pakistan). Chitosan (LMW) (PubChem CID: 21896651) and sodium tripolyphosphate (STPP) (PubChem CID: 24455) were purchased from Sigma-Aldrich (Steinheim, Germany). Carrageenan (κ-kappa) (PubChem CID: 11966249) was from CP Kelco, a Huber company. Carbopol 940, ethanol, triethanolamine, and sodium hydroxide were obtained from Sigma-Aldrich (Steinheim, Germany). 

### 3.2. Methods

#### 3.2.1. Design of Experiment (Box–Behnken Design)

For the evaluation of the influence of different formulation variables on entrapment efficiency, particle size, and drug permeation, the response surface methodology (RSM) was used as a statistical tool and mathematical technique. A 3-factor, 3-level Box–Behnken design (BBD) was used for designing 17 experimental runs (25). Two different formulations were also prepared, one without CRG (N18) and one without STPP (N19), to study the effect of these polymers (Table 8). Carrageenan concentration (X1), drug concentration (X2), and stirring speed (X3) were selected as independent process variables, each with three levels: [X1 (0.05%, 0.06%, 0.07%), X2 (30%, 40%, 50%), and X3 (600, 750, and 900 rpm)]. The effect of change in independent variables on dependent variables (i.e., entrapment efficiency (Y1), particle size (Y2), and percentage of cumulative drug permeation (Y3)) was studied using one-way analysis of variance (ANOVA) using Stat-Ease (Design-Expert 9.0.6.2). The following non-linear quadratic equation (Equation (4)) was used to evaluate the significance of each independent variable.
Y = β0 + β1X1 + β2X2 + β3X3 + β11X21 + β 22X22 + β33X23 + β12X1X2 + β13X1X3 + β23X2X3 (4)
where:Y is the dependent variable;X1, X2, and X3 are independent variables;β1, β2, and β3 are non-linear coefficients;β11, β22, and β33 are squares of coefficients;β12, β13, and β23 are the interaction coefficients of this non-linear equation.

#### 3.2.2. Experimental Method

In two main steps, nanoparticulate-containing Carbopol gel was formulated.

(A.)Preparation of NAP-loaded CS/CRG nanoparticles

NAP-loaded CS/CRG nanoparticles were fabricated by a polyelectrolyte complexation technique with minor alterations to a previously reported methodology [24,41]. Furthermore, ionotropic gelation was performed in the presence of a counter ion, i.e., STPP [61]. 

CS solution (0.1%) was prepared by mixing CS in 1% acetic acid (*v*/*v*) and stirring overnight at 40 °C using a magnetic stirrer. Any undissolved chitosan was removed by filtration. CRG was mixed in distilled water (10 mL) at 60 °C using a magnetic stirrer. To fabricate the drug-loaded CS/CRG complex nanoparticles, NAP was mixed in ethanol (2 mL) and added to the CRG solution. NAP concentration in the CRG solution was kept in such a range so that nanoparticles were prepared with 30%, 40%, and 50% (*w*/*w*) of drug concentration in the CS solution. Polyelectrolyte complex formation occurred when drug-containing CRG solutions were incorporated into a CS solution under magnetic stirring for 60 min. Then, CS was polymerized through ionic gelation with STPP. An STTP solution of 0.5% (*w*/*v*) was used as a crosslinker, prepared by dissolution in distilled water. The STPP solution was added dropwise into the drug-containing CS solution under magnetic stirring for 1 h. Freshly prepared nanoparticles were then subjected to centrifugation (Sigma 1-14; Sigma Laborzentrifugen GmbH, Osterode am Harz, Germany) at 12,000 rpm for 40 min. The supernatants were used to find the entrapped drug (% EE), and nanoparticles were resuspended in 1 mL of purified water and dried by lyophilization for 48 h.

(B.)Formation of Carbopol 940 gel containing naproxen-loaded CS/CRG nanoparticles

Carbopol 940 (0.1%) was used as a gelling agent. Ca-940 was mixed in distilled water by continuous stirring at 800 rpm for 1 h by homogenizer (VELP Scientifica, Usmate Velate, Italy) to form Ca-940 solutions. Gel pH (6.2–6.8) was maintained by using 0.05% triethanolamine. Optimal viscosity, clear visual appeal, compatibility, and spreadability of a gelling agent with polymers and drugs were the core elements in selecting the gelling agent. Then, 50 mg NAP-containing nanoparticles were accurately measured and mixed thoroughly with the above-mentioned Ca-940 solutions [37]. 

### 3.3. Characterization of Nanoparticles

#### 3.3.1. Entrapment Efficiency

The entrapment efficiency (EE) of the drug in the CS/CRG nanoparticle complex was evaluated using both direct and indirect methods [62,63]. Moreover, when the results of both methods were compared, the results were almost the same. In this study, we used an indirect method for the study’s convenience, and results were produced for 19 formulations in the manuscript. As described in the methodology, drug incorporation evaluation of naproxen-loaded nanoparticles was performed via centrifugation at 12,000 rpm for 40 min. The supernatant containing unentrapped drug was separated and added into the water obtained after washing. By combining both solutions, the concentration was calculated by taking 3-times UV absorbance at 262.6 nm [62]. The percentage entrapment efficiency was calculated by using the equation given below.
% E.E. = (Total amount of drug − Unentrapped drug)/(Total amount of drug) × 100 (5)

#### 3.3.2. Particle Size, Polydispersity Index, and Zeta Potential Determination

The particle size, zeta potential, and PDI were determined via the dynamic light scattering (DLS) technique (Zetasizer Nano ZS90, Malvern Panalytical, Malvern, UK). The dried lyophilized nanoparticle powder samples were suspended in distilled water and vortexed before measurement to prevent clumping. Then, a 1-mL nanoparticle dispersion sample was taken and diluted 10 times with deionized water in Zetasizer cuvettes for particle size and zeta potential determination. 

#### 3.3.3. Scanning Electron Microscopy (SEM)

The nanoparticles’ morphology was inspected by scanning electron microscopy (SEM) with a Philips XL30 scanning microscope (JSM-IT-100, JEOL, Japan) at an accelerating 5–10 kV voltage. Before the assessment, NP samples were placed on an electroconductive chip of silicon on top of aluminum stubs, under an argon atmosphere (JSM-5910, JEOL Ltd., Tokyo, Japan) and through a sputter coater. Both formulations (“lyophilized” and “water dispersion”) were selected to study the morphological characteristics and appearance. Photomicrographs of coated nanoparticles were taken to reveal their external surface and morphological characteristics [64].

#### 3.3.4. Fourier Transforms Infrared Spectroscopy (FTIR) 

Fourier transform infrared (FTIR) spectroscopic analysis was used to study the drug–polymer interaction, preferably for compatibility studies [65]. Pure NAP, chitosan (LMW), carrageenan (kappa), drug–polymer physical mixture, and other formulations were analyzed and compared to find any possible interaction between components of the formulation by FTIR (Tensor 27 IR; Bruker, Karlsruhe, Germany). The physical mixture was prepared by geometric mixing of the components in a mortar for 5 min and then “sieving through 100 mesh size sieves” [66]. Powdered nanoparticles sample were placed on ATR crystal and pressed on the crystal’s face by rotating and turning the arm to achieve efficient contact. Scanning was done for 16 s in the range of 4000–500 cm^−1^.

#### 3.3.5. Powdered X-ray Diffraction (pXRD) Analysis

Powdered X-ray diffraction (XRD) analysis is an effective tool to analyze the amorphous and crystalline nature of particles and biopolymer electrolytes. The powdered X-ray diffractometry (PXRD) of NAP, CS, CRG, physical mixture, and optimized formulation (No) was recorded by a powder X-ray Diffractometer (JDX 3532; JEOL Ltd., Tokyo, Japan) to analyze the solid-state stability of the components mentioned above. Samples were examined at 2θ in the range of 10° to 60°. The sample’s X-ray diffraction patterns were detected using the Cu line as a source of radiation, and a power of 35 kV with a 40 mA current was supplied.

#### 3.3.6. Acute Oral Toxicity Study

As per the guidelines of the Organization for Economic Co-operation and Development (OECD), the toxicity study was designed for polyelectrolyte complex nanoparticles to evaluate the fabricated particles’ safety and biocompatibility. According to the guidelines, the animal house environment was maintained, i.e., 40% relative humidity, and 22 °C ± 3 °C room temperature with a sequence of light and dark cycles. Rabbits were selected as the animal model due to data availability and the recognized pathophysiology; ultimately, the effects on human health can be predicted. All the rabbits were assigned into Group I and Group II, each group having six rabbits (*n* = 6) that were housed in ventilated and cleaned cages. Standard food and water were administered to Group I and taken as the Control group, whereas for Group II (Treatment group), treatment therapy was administered. The toxicity studies were conducted after approval (08-2020/PAEC) from the Pharmacy Animal Ethics Committee (PAEC), The Islamia University of Bahawalpur. Animals in both groups were keenly observed for 15 days, and then samples were collected for the evaluation of blood chemistry [67].

#### 3.3.7. Characterization of Naproxen-Loaded, CS/CRG-Nanoparticle-Containing Ca-940 Gel 

The above-formulated NAP-loaded CS/CRG-nanoparticle-containing Ca-940 gel was subjected to evaluation for the following parameters:

##### Appearance, pH, Viscosity, and Spreadability

All prepared NAP nanoparticle-containing gel preparations were studied for their color, clarity, homogeneity, and presence of lumps by visual inspection [48]. A digital pH meter (inoLab; Xylem Analytics, Germany) was used for the determination of pH of the Ca-940 gel formulations. NAP nanoparticle-containing gels were transferred to a graduated beaker and made a 50-milliliter final volume with distilled water. The freshly prepared gel formulations were measured by using a digital pH meter immersed entirely into the gel system until a constant reading was achieved. Each formulation’s pH was measured in triplicate by calculating the average [68,69]. The viscosity of the NAP gel was measured at 25 °C using a Brookfield RST Cone Plate Rheometer with spindle CP 62 at 4 rpm for 50 s [70]. 

Spreadability is one of the most critical parameters of which to evaluate the ideal quality, as the spreading value is critical for the gel formulation’s therapeutic efficacy. Briefly, 0.5 g of gel formulation was placed on a glass plate within a pre-marked circle with a diameter of 1 cm, over which a second glass plate was placed to measure the spreadability of the NAP-loaded gel. A weight of 100 g was allowed to rest on the upper glass plate for 5 min. The gel formulations’ spreadability was measured in triplicate by calculating an increase in diameter [52].

##### Drug Content

A total of 100 mL of ethanolic phosphate buffer (EPB) was prepared, and a sample of gel containing NAP nanoparticles (100 mg) was dispersed in it. For the complete solubilization of the drug into EPB, a mechanical shaker (IKA-Werke, Germany) was used for 4 h. After filtration, the UV absorbance of the sample was taken at 262.6 nm by using a phosphate buffer (pH 7.4) as a blank [71].

##### Skin Irritation Studies for Naproxen-Loaded Gel 

For the evaluation of the skin irritation potential of NAP nanoparticle-containing gels, the Draize patch test was performed. Albino Wistar rats (200 ± 0.25 g) of any gender were supplied by Islamia University Animal Research Centre Bahawalpur, Pakistan, and were individually housed in the animal house with a supply of food and water. Twenty rats were separated into two groups (*n* = 2): group 1 received all NAP formulations containing nanoparticle gel, and group 2 received pure NAP (control gel). A precise area on the back of the rats was shaven 24 h before the formulation application. A required amount of gel (equal to 5 mg naproxen) was applied to the rats’ hair-free skin. The test site remained intact for 48 h, then the gel was detached, and the resulting skin reactions were observed on 48th, 72nd, and 96th hour. An erythema score was given from 0 to 4, conditional on the degree of erythema [72].

##### Stability Studies

Stability studies (accelerated stability studies) for NAP nanoparticle-containing gels were performed by placing the formulations at different stability conditions. The formulations of nanoparticles were divided into four batches; one batch was kept at room temperature 27 ± 1 °C, the second at 4 ± 1 °C, the third at 37 ± 1 °C, and the fourth at 45 ± 1 °C for 3 months. At weekly intervals, the absorbance of samples was taken at 262.6 nm using a phosphate buffer (pH-7.4), and physicochemical properties were estimated [73].

### 3.4. Evaluation of Ex Vivo Drug Permeation Study

#### 3.4.1. Preparation of Full-Thickness (FT) Rat Skin

The experiment was approved by the Pharmacy Research Ethics Committee and conducted according to protocol. The skin was obtained from the albino rat (weight: 200–400 g). An FT rat skin was taken out by removing the fat using a surgical scalpel, adhering to the dermis side of the skin. As a final point, the skin was washed with a phosphate buffer (7.4), kept in aluminum foil, stored at −20 °C, and was used within a week [74].

#### 3.4.2. Ex Vivo Permeation Studies by Using Franz Diffusion Cells 

Franz diffusion cells were used to perform the skin permeation studies for 48 h by using FT rat skin. The skin sample was kept in a phosphate buffer solution for 2 h before the experiment provided optimal skin hydration. The receptor chamber was filled with a phosphate buffer of pH 7.4, and the FT rat skin was mounted on a diffusion cell. The whole assembly was kept over a magnetic stirrer, and the temperature was maintained at 32 ± 0.5 °C. After maintaining the skin, NAP nanoparticle-containing gel equal to 5 mg NAP was spread over the skin and covered to avoid solvent evaporation. At a specific interval, a sample of 200 µL was withdrawn from the lower compartment and was replaced with fresh medium. UV spectrophotometric analyses were performed at 262.6 nm [75]. The ex vivo drug permeation of optimized nanoparticle gel (No) was compared with the control gel using ANOVA. The control gel was prepared by adding pure drug (NAP 50 mg) into the Carbopol 940 gel (100 mg). Additionally, 10 mg of control gel (containing 5 mg of NAP) was used for experimental purposes and compared with the optimized nanoparticle-containing gel equal to 5 mg of NAP. Drug permeation was evaluated by using the following formula:Percent drug permeated = (Quantity of drug permeated at time ‘)/(quantity of drug-loaded in nanoparticles) ×100 (6)

The percentage of drug permeation through FT rat skin was plotted between the time and flux (penetration rate), which was determined.

### 3.5. In Vivo Anti-Inflammatory Studies in Rats

#### 3.5.1. Animals

Albino rats of 150 + 10 g were bred in the animal house of the Pharmacy department of The Islamia University of Bahawalpur, Pakistan, and were fed on a standard laboratory diet with water. Polypropylene cages were used for rats by maintaining the room temperature at 25 + 1 °C, a photoperiod (12:12 h) light and dark cycle, and humidity at 55–60%. The animal trial studies were approved by the Pharmacy Animal Ethics Committee (08-2020/PAEC) at The Islamia University of Bahawalpur.

#### 3.5.2. Treatment Protocol

Animals were distributed into 4 groups (6 each) and treated accordingly: normal control, formalin, formalin + NAP NP gel (15 mg/kg), and formalin + NAP control gel. Inflammation was induced by administering intradermal formalin (0.1 mL of 2% *v*/*v*) in each rat’s right hind paw on the first and third days of the experiment. A paw edema meter was used to measure the paw volume 1, 3, and 5 h after treatment. For the calculation of the degree of swelling in the paw and the inhibition rate of edema, the following formula was used
Percentage inhibition = 1 − VT/VC × 100(7)
where VT and VC are the paw volume of the treatment and control group, respectively. The rats were sacrificed by decapitation at the end of the 7 days. After that, a histopathological evaluation of the samples was performed.

### 3.6. Histopathology Analysis

The right paw tissues were removed, fixed in 10% formalin, and handled with paraffin embedding. Tissue samples of 3 mm thickness were selected and placed on the slides. Tissues stained with hematoxylin for investigating under a light microscope for histopathological changes [76].

### 3.7. Statistical Analysis 

Statistical analysis was applied to the data in order to evaluate the influence of independent variables on the response variables. Design-Expert software (9.0.6.2) was selected as a statistical tool to perform statistical analysis. For the evaluation of the significance of each of the independent variables, an ANOVA test was applied. A value of “*p*” less than 0.05 was considered significant.

### 3.8. Model Dependent Permeation Kinetic Analysis

For the determination of the order and mechanism of drug permeation, different kinetic models were applied to ex vivo permeation data. Regression analysis was applied to the ex vivo permeation data, and a coefficient of zero-order as the cumulative amount of drug permeates vs. time [77], first-order as the cumulative log percentage of drug remaining vs. time [78], Higuchi as the cumulative percentage of drug permeated vs. square root of time [79], and Korsmeyer–Peppas [80] models were determined, respectively. They were compared to describe the order and mechanism of drug permeation. The value of diffusion exponent “*n*” was estimated by fitting the permeation data to the Korsmeyer–Peppas model, and the mechanism of drug permeation was determined. If the value of *n* is 0.45, then the formulation follows the Fickian diffusion. If the value of *n* is between 0.45 and 0.88, then the formulation follows an anomalous (non-Fickian) diffusion. If the value of *n* is 0.89, then permeation is case II transport, and if the value of *n* is greater than 0.89, then the formulation follows super case II transport [81].

## 4. Conclusions

This study demonstrated the successful preparation of CS/CRG-based, NAP nanoparticle-containing gel to be used as a sustained TDDS. NAP-loaded, CS/CRG-nanoparticles were successfully synthesized using optimized variables: polymer concentration, drug concentration, and stirring speed proposed by BBD. As a novel approach to nanoparticle-containing gel as a TDDS, this study was initiated to design a polymeric nanoparticle-containing gel using biodegradable polymers through the polyelectrolyte complexation method. Statistical analysis displayed that high polymer concentration (CRG 0.07%) and stirring speed (900 rpm) but low drug concentration (30%) are mandatory for additionally sustained permeation. NAP-loaded CS/CRG nanoparticle-containing Carbopol 940 gel showed sustained permeation of NAP for a period of 48 h in an ex vivo permeation study using FT rat skin. Ex vivo drug permeation studies verify the sustained liberation of NAP from CS/CRG nanoparticles for 48 h following a non-Fickian diffusion mechanism and zero-order release kinetics. It has been observed that an optimized batch produced a gel with good consistency, homogeneity, spreadability, stability, and enhanced permeation as compared to the control gel. The acute oral toxicity studies show no signs of incompatibility and toxicity in the developed nanoparticles in histopathological slides. In vivo anti-inflammatory studies confirmed ex vivo study results through an increased inhibition percentage of paw edema and a marked reduction in inflammation markers through histopathological illustrations. The values of the outcome variables were very close to the predicted variables by BBD, which signifies the robustness and reliability of BBD. Accordingly, it is expected that novel NAP-loaded nanoparticle gels can be exploited as a beneficial substitute to treat arthritis via improved permeation profile and better patient compliance.

## Figures and Tables

**Figure 1 pharmaceuticals-14-00557-f001:**
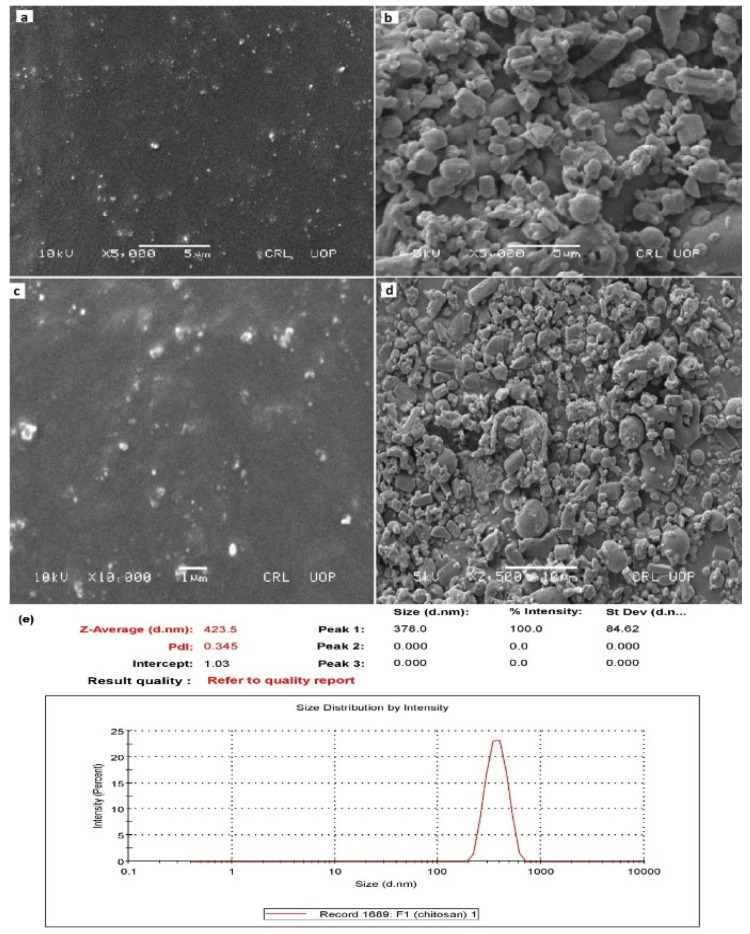
(**a**,**c**) SEM microphotographs of N11 and optimized No naproxen-loaded CS/CRG nanoparticle formulation liquid dispersion; (**b**,**d**) lyophilized; (**e**) and particle size and PDI of N15 formulation.

**Figure 2 pharmaceuticals-14-00557-f002:**
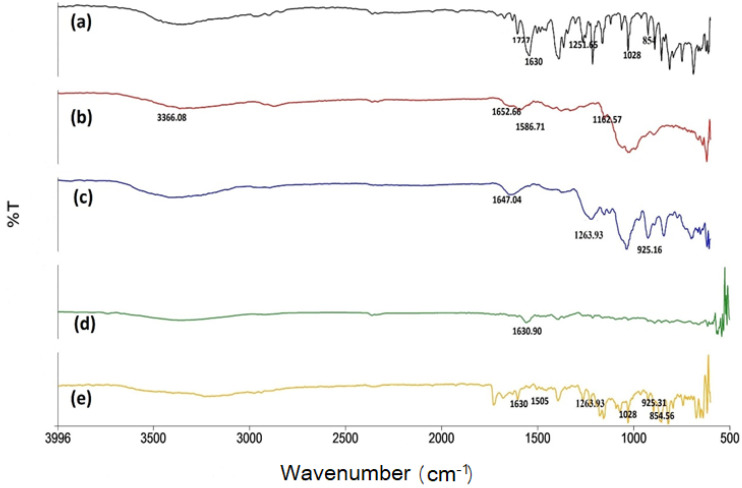
(**a**) FTIR spectra of naproxen, (**b**) chitosan, (**c**) carrageenan, (**d**) physical mixture, and (**e**) optimized formulation.

**Figure 3 pharmaceuticals-14-00557-f003:**
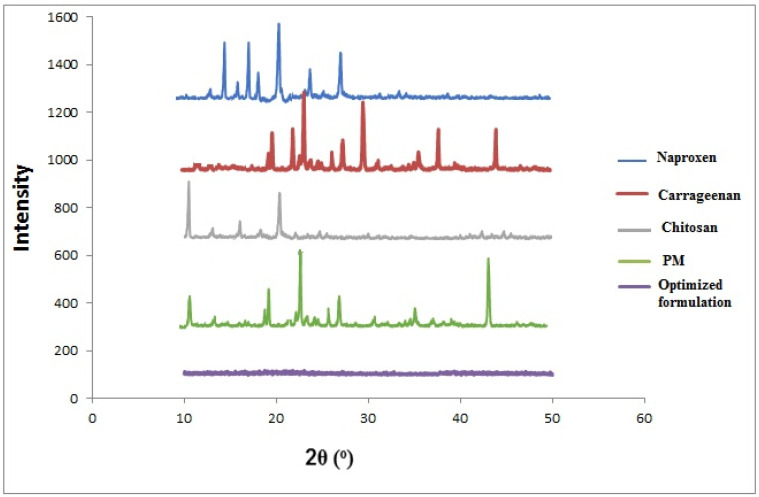
XRD spectra of naproxen, carrageenan, chitosan, physical mixture, and optimized formulation.

**Figure 4 pharmaceuticals-14-00557-f004:**
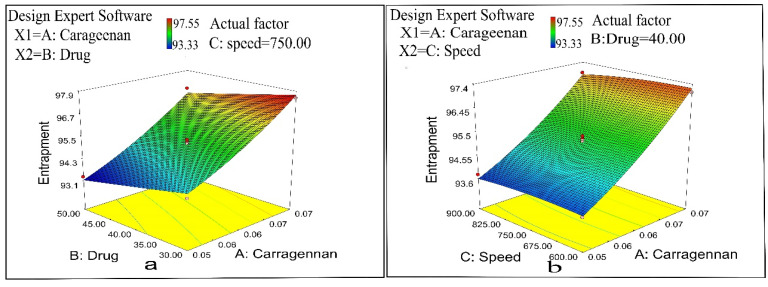
(**a**) Influence of polymer concentration and drug concentration on entrapment efficiency and (**b**) influence of polymer concentration and stirring speed on entrapment efficiency.

**Figure 5 pharmaceuticals-14-00557-f005:**
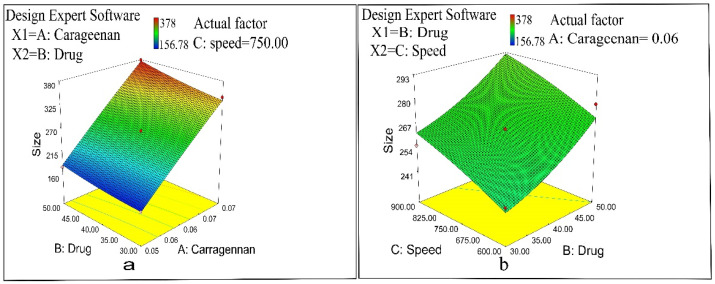
(**a**) Influence of polymer concentration and drug concentration on particle size and (**b**) influence of drug concentration and speed on particle size.

**Figure 6 pharmaceuticals-14-00557-f006:**
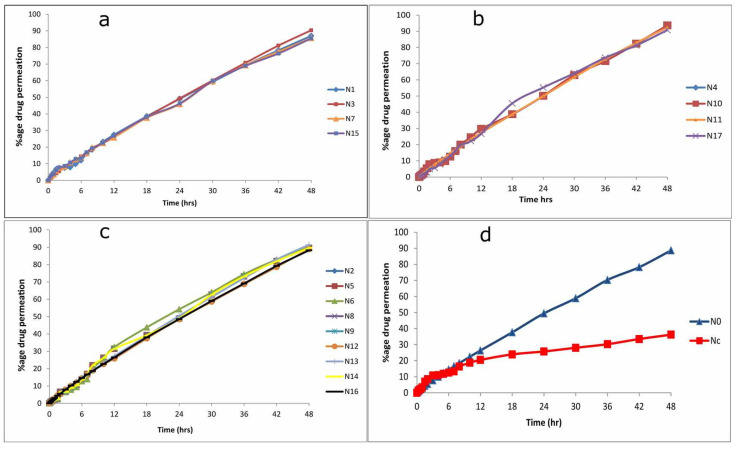
(**a**) Drug permeation percentage of formulation with 0.05% of carrageenan; (**b**) with 0.06% carrageenan; (**c**) with 0.07% carrageenan; (**d**) optimized formulation (No) containing Carbopol 940 gel and control formulation (Nc) containing Carbopol 940 gel through FT rat skin.

**Figure 7 pharmaceuticals-14-00557-f007:**
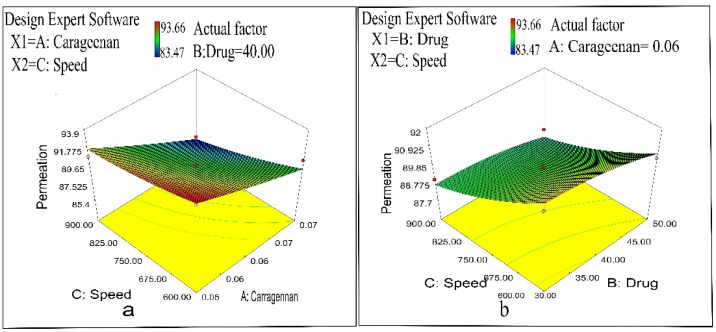
(**a**) Influence of polymer concentration and stirring speed on drug permeation percentage. (**b**) Effect of drug concentration and stirring speed on drug permeation percentage.

**Figure 8 pharmaceuticals-14-00557-f008:**
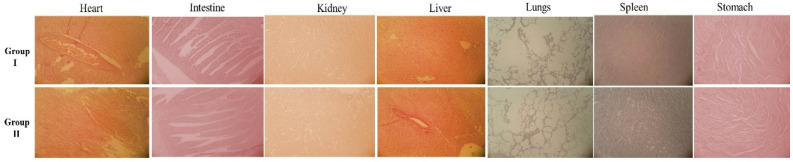
Histopathological examination of different organs of Group I (control group) and Group II (treatment group) of rabbits.

**Figure 9 pharmaceuticals-14-00557-f009:**
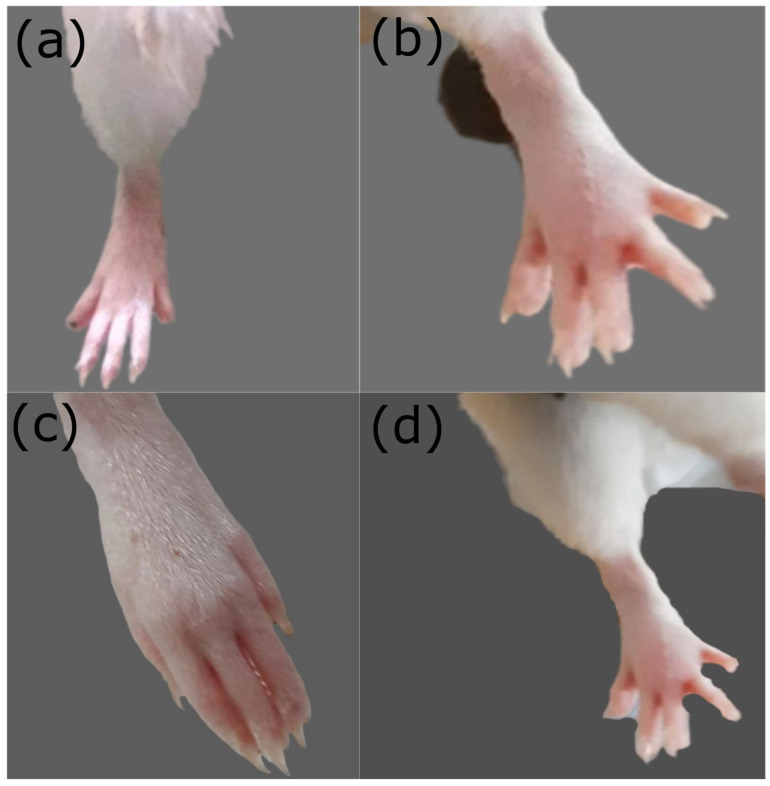
(**a**) Photograph of control animal rat paw; (**b**) photograph of formalin-induced rat paw; (**c**) photograph of NAP-gel-treated rat paw; (**d**) photograph of control-gel-treated rat paw.

**Figure 10 pharmaceuticals-14-00557-f010:**
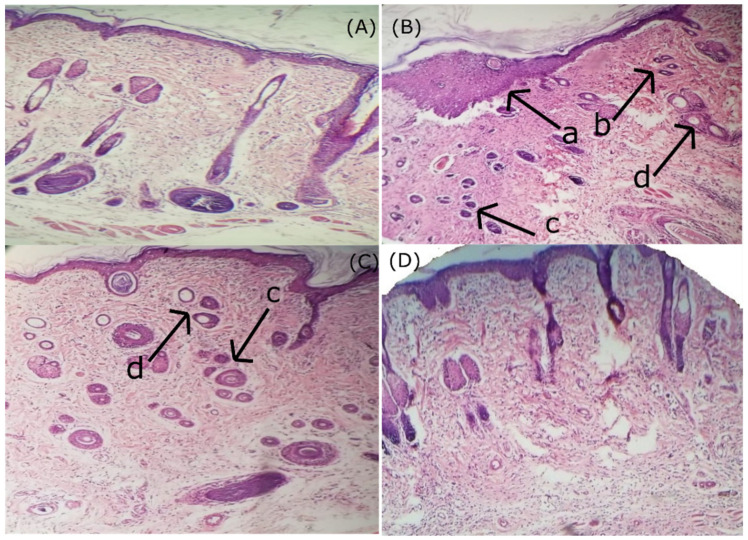
Histopathological study of rat paw tissue; (**A**) control paw tissue showed normal epidermis, deep dermis, and subcutaneous tissues; (**B**) (a) group treated with formalin alone showed marked hyperkeratosis of skin with epithelial proliferation. (b) Under dermis loosens and is edematous. (c) Deep dermis and subcutaneous tissues show moderate inflammatory cell infiltration. (d) edema, and proliferation of collagenous tissue. (**C**) (c) FIE + control-gel-treated group shows deep dermis and subcutaneous tissues with mild inflammatory cell infiltration, (d) edema, and proliferation of collagenous tissue. (**D**) FIE + NAP-gel-treated group showed a marked reduction in the injury to paw tissue. Most of the histological changes were minimized and found negligible as compared to the group treated with formalin alone. (FIE = formalin-induced edema).

**Table 1 pharmaceuticals-14-00557-t001:** Effect of formulation components and process variables on different characteristics.

Formulation Code	Entrapment Efficiency (%)	Particle Size (nm)	PDI	Zeta Potential (mV)	% Cumulative Drug Permeation	% Yield
N1	97.55%	342.45 ± 16.33	0.308	+25 ± 0.9	86.98	71.76%
N2	95.12%	264.33 ± 35.78	0.422	+29 ± 1.3	89.66	70.46%
N3	97.08%	332.29 ± 8.43	0.318	+26 ± 1.9	90.34	72.48%
N4	94.12%	162.75 ± 22.43	0.510	+34 ± 1.1	93.33	67.01%
N5	95.25%	264.33 ± 35.78	0.422	+34 ± 1.6	89.66	70.54%
N6	94.66%	277.54 ± 6.74	0.443	+33 ± 3.1	90.25	71.61%
N7	96.94%	362.43 ± 54.64	0.355	+27 ± 2.6	85.66	72.72%
N8	95.36%	264.33 ± 35.78	0.422	+33 ± 3.1	89.66	70.22%
N9	95.08%	264.33 ± 35.78	0.422	+34 ± 1.1	89.66	70.72%
N10	93.66%	156.78 ± 45.53	0.504	+35 ± 2.1	93.66	67.26%
N11	93.33%	176.68 ± 27.03	0.512	+33 ± 0.8	92.34	67.96%
N12	95.77%	255.38 ± 18.43	0.465	+30 ± 1.3	88.93	69.92%
N13	96.03%	243.22 ± 10.55	0.412	+29 ± 0.6	91.45	69.72%
N14	95.18%	264.33 ± 35.78	0.422	+33 ± 3.1	89.66	70.60%
N15	96.86%	378.0 ± 84.62	0.345	+28 ± 2.3	85.47	73.96%
N16	94.01%	290.08 ± 24.22	0.454	+31 ± 0.4	88.23	69.53%
N17	93.78%	188.32 ± 16.69	0.521	+33 ± 0.8	90.76	67.46%
N18	90.22%	407.20 ± 84.62	0.533	+26 ± 2.1	76.80	53.2%
N19	89.34%	674.20 ± 47.48	0.736	+12 ± 2.2	63.34	28%
No (optimized)	95.26%	355.7 ± 79.8	0.381	+25 ± 3.1	88.66	70.78%

**Table 2 pharmaceuticals-14-00557-t002:** ANOVA test values of dependent response variables.

Sr. No.	Response Variables	Statistical Term	*p*-Value
1.	Entrapment efficiency	Polymer concentration	0.0001
		Drug concentration	0.0001
		Stirring speed	0.2258
2.	Particle size	Polymer concentration	0.0001
		Drug concentration	0.0005
		Stirring speed	0. 0033
3.	% cumulative drug permeation	Polymer concentration	0.0001
		Drug concentration	0.0810
		Stirring speed	0. 0008

**Table 3 pharmaceuticals-14-00557-t003:** Evaluation of naproxen-loaded, CS/CRG-nanoparticle-containing gel.

Formulation Code	Appearance and Homogeneity	pH	Viscosity (cps)	Spreadability (cm)	Drug Content (%)	Skin Irritation Score
N1	+++	6.8	5450	3.2	92	0
N2	+++	6.4	5325	3.6	89	1
N3	+++	6.7	5455	2.8	94	0
N4	+++	6.8	5272	4.9	87	0
N5	+++	6.4	5328	3.5	88	1
N6	+++	6.5	5335	2.4	92	1
N7	+++	6.8	5458	2.6	93	0
N8	+++	6.4	5321	3.6	90	1
N9	+++	6.4	5332	3.7	90	1
N10	+++	6.2	5283	4.6	87	1
N11	+++	6.2	5296	4.4	88	1
N12	+++	6.3	5340	3.3	89	1
N13	+++	6.4	5338	3.4	90	1
N14	+++	6.4	5330	3.5	93	1
N15	+++	6.6	5466	2.9	95	0
N16	+++	6.3	5345	3.7	92	1
N17	+++	6.2	5218	4.2	91	1
N18	+++	6.4	3200	2.6	85	1
N19	+++	6.3	5102	2.2	87	1
No	+++	6.5	5040	2.3	93	0

+++ Excellent.

**Table 4 pharmaceuticals-14-00557-t004:** Kinetic parameters of ex vivo permeation studies.

Code	Zero-Order		First-Order		Higuchi Model		Korsmeyer–Peppas Model	
	R^2^	K	R^2^	K	R^2^	K_H_	R^2^	*n*
N1	0.9899	1.922	0.9818	0.030	0.8877	10.130	0.9976	0.869
N2	0.9872	2.008	0.9808	0.033	0.8890	10.594	0.9963	0.859
N3	0.9942	1.971	0.9782	0.031	0.8844	10.365	0.9999	0.884
N4	0.9921	2.020	0.9741	0.033	0.8834	10.629	0.9980	0.883
N5	0.9872	2.008	0.9808	0.033	0.8890	10.594	0.9963	0.859
N6	0.9851	2.049	0.9800	0.034	0.8824	10.790	0.9935	0.865
N7	0.9877	1.901	0.9833	0.030	0.8979	10.065	0.9986	0.846
N8	0.9872	2.008	0.9808	0.033	0.8890	10.594	0.9963	0.859
N9	0.9872	2.008	0.9808	0.033	0.8890	10.594	0.9963	0.859
N10	0.9922	2.025	0.9739	0.033	0.8830	10.649	0.9980	0.884
N11	0.9947	2.015	0.9753	0.032	0.8825	10.586	0.9999	0.889
N12	0.9933	1.928	0.9809	0.031	0.8872	10.151	0.9999	0.877
N13	0.9938	2.008	0.9773	0.032	0.8853	10.562	0.9999	0.881
N14	0.9872	2.008	0.9808	0.033	0.8890	10.594	0.9963	0.859
N15	0.9859	1.897	0.9853	0.030	0.9018	10.060	0.9986	0.835
N16	0.9923	1.933	0.9822	0.031	0.8903	10.191	0.9999	0.868
N17	0.9876	2.036	0.9786	0.033	0.8774	10.693	0.9941	0.880
N18	0.9996	1.625	0.9779	0.023	0.8519	8.433	0.9998	0.976
N19	0.9981	1.366	0.9859	0.018	0.8451	7.100	0.9983	0.977
No	0.9927	1.936	0.9818	0.031	0.8884	10.197	0.9998	0.873

**Table 5 pharmaceuticals-14-00557-t005:** Different biochemical, hematological, and weight variation studies in Group I (control) and Group II (treatment).

Parameter/Test	Group I (Control)	Group II (Treatment)
Biochemical Parameters		
AST/SGOT (IU/L)	144.23 ± 2.0	146.02 ± 2.50
Creatinine (mg/dL)	0.75 ± 0.11	0.88 ± 0.09
Triglycerides (mg/dL)	57 ± 3.11	56 ± 2.04
Total cholesterol (mg/dL)	62.31 ± 3.76	60.08 ± 5.10
Serum uric acid (mg/dL)	3.21 ± 0.02	3.43 ± 0.03
Serum urea (mg/dL)	12.56 ± 3.04	14.76 ± 2.32
Hematological Parameters		
Hemoglobin Hb (g/dL)	13.21 ± 0.32	13.48 ± 0.41
Red blood cells (RBCs) × 10^6^/mm^3^	6.12 ± 0.51	5.66 ± 0.61
White blood cells (WBCs) × 10^9^/L	6.62 ± 0.02	6.77 ± 0.40
Platelets × 10^9^/L	4.05 ± 2.05	4.16 ± 2.06
Neutrophils (%)	55.80 ± 4.05	57.55 ± 5.11
Lymphocytes (%)	38.30 ± 1.08	39.21 ± 1.02
Monocytes (%)	3.60 ± 0.11	3.65 ± 0.21
Mean corpuscular volume (%)	83.66 ± 2.10	84.84 ± 2.40
Mean corpuscular hemoglobin (pg/cells)	23 ± 3.05	24 ± 2.25
Mean corpuscular hemoglobin concentration (%)	33.30 ± 2.21	33.92 ± 1.41
Rabbit Organ Weights		
Heart	4.33 ± 0.20	4.40 ± 0.16
Kidney	12.33 ± 0.21	13.75 ± 0.81
Liver	7.22 ± 2.01	8.22 ± 2.11
Lungs	9.22 ± 0.38	9.25 ± 0.51
Spleen	1.12 ± 0.11	1.13 ± 0.14
Stomach	12.22 ± 0.41	13.01 ± 0.70

Note. All values are expressed as mean ± *SD* (*n* = 3).

**Table 6 pharmaceuticals-14-00557-t006:** Evaluation of optimized formulation of naproxen-loaded CS/CRG nanoparticles and stability study of optimized formulation.

**Optimized Formulation (No)**					
**Parameters**	**Entrapment Efficiency (%)**	**Particle Size (nm)**	**PDI**	**Zeta Potential (mV)**	**Cumulative Drug Permeation (%)**
Predicted variables by design expert	97.43	345.015	-	-	85.7359
Experimental values	95.26 ± 3.23	355.7	0.381	+25 ± 3.1	88.66
**Stability study of optimized formulation**					
**Time**	**Appearance**	**pH**	**Viscosity (cps)**	**Spreadability (cm)**	**Drug content (%)**
Day 0	++	6.78	5483	2.82	94
Day 30	++	6.74	5496	2.87	92
Day 60	++	6.76	5509	3.20	90
Day 90	++	6.66	5483	3.27	88

++ Good.

**Table 7 pharmaceuticals-14-00557-t007:** Percentage inhibition of paw edema after 1, 3, and 5 h on experimental animals.

Percentage Inhibition of Paw Edema			
Groups	1 h	3 h	5 h
FIE	16	16	16
Formalin + NAP NP gel	36 ± 0.96	57 ± 0.79	79 ± 0.85
Formalin + NAP control gel	21 ± 0.76	45 ± 1.13	59 ± 1.01

**Table 8 pharmaceuticals-14-00557-t008:** Experimental parameters based on a Box–Behnken design.

Formulation Code	Naproxen (%)	Chitosan (%)	Carrageenan (%)	STPP (%)	Stirring Time (h)	Stirring Speed (rpm)
N1	30	0.1	0.07	0.5	1	750
N2	40	0.1	0.06	0.5	1	750
N3	40	0.1	0.07	0.5	1	600
N4	30	0.1	0.05	0.5	1	750
N5	40	0.1	0.06	0.5	1	750
N6	50	0.1	0.06	0.5	1	600
N7	40	0.1	0.07	0.5	1	900
N8	40	0.1	0.06	0.5	1	750
N9	40	0.1	0.06	0.5	1	750
N10	40	0.1	0.05	0.5	1	600
N11	50	0.1	0.05	0.5	1	750
N12	30	0.1	0.06	0.5	1	900
N13	30	0.1	0.06	0.5	1	600
N14	40	0.1	0.06	0.5	1	750
N15	50	0.1	0.07	0.5	1	750
N16	50	0.1	0.06	0.5	1	900
N17	40	0.1	0.05	0.5	1	900
N18	40	0.1	0	0.5	1	750
N19	40	0.1	0.06	0	1	750
No (optimized)	30	0.1	0.07	0.5	1	900

## Data Availability

Data is contained within the article.

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
