# Peer review of "Optimization of Novel Naproxen-Loaded Chitosan/Carrageenan Nanocarrier-Based Gel for Topical Delivery: Ex Vivo, Histopathological, and In Vivo Evaluation"

_pharmaceuticals, 2021, doi:10.3390/ph14060557_

Round 1
Reviewer 1 Report
The manuscript by Sobia Noreen et al. entitled “Optimization of novel Naproxen loaded chitosan/ carrageenan nanocarrier based gel for topical delivery: ex vivo, histopathological and in vivo evaluation” reports formulation, characterization and ex vivo and in vivo evaluation of a Carbopol gel containing nanoparticles prepared by polyelectrolyte complexation method with Naproxene as drug. The introduction is complete and clearly presents the background required to fully understand the manuscript. Materials and Methods are well explicative, and results seems in accordance with the aim of the manuscript. Conclusions are well supported by the obtained results. English need to be revised and many typos need to be corrected. Therefore, in my opinion some parts need to be clarified as follows:
Line 20: The sentence is misleading because the verb is missing. Please check and fix it.
Line 44: I suppose “NF-α” is a typo for “TNF-α”, please correct it.
Line 70: This sentence “Typically, PEC is the formation of the complex in a reaction among a polyanion (PA- ) with counter cations (CC+), a polycation (PC+) with counter anions (CA-)” seems out of context written in this way, in my opinion it would fix better if it were moved below after description of Chitosan, for example after line 83.
Line 112: Description about Particle size and Zeta potential of nanoparticles N0 and N11 is missing. Since the Authors performed even these analyses, I suggest adding the information to confirm the data obtained from SEM imaging.
Line 137: Through the whole manuscript the “carrageenan” was reported in different way as “?-carrageenan”, “carrageenan kappa” or “CRG”, in my opinion it would be better choose a single denomination and standardize it for the whole text.
Line 249: according to this part the ex vivo permeation study was performed for 48 hours, but there is not reference about that in the respective “materials and methods” paragraph. Thus, I strongly suggest adding the missing information in the 4.4.2 section.
Line 250: Authors refer to formulation using “F1-F19” here but in the table 1 they instead use “N1-N19”. Please check and correct it.
Line 314: Table 3 I suppose it would be necessary adding a legend to explain what the “+++” refer to in terms of appearance and homogeneity.
Line 329: this section could be improved, since the study on kinetics is very important in the pharmaceutical field, I suggest to add some information about n-value and linear regression coefficient experimentally obtained for all the formulations. Therefore, adding a summary graph could help to better understand the data.
Lines 350-352: Even though I understood what the Authors want to demonstrate performing oral toxicity, this statement could be misunderstood because the route of administration could modify the toxicity (i.e. after oral administration the first pass effect or stomach/intestine pH could affect the bioavailability or chemical structure of the drug/excipients). Thus, I strongly suggest revising this part adding some references.
Line 532: The name of the centrifuge is missing, anyway I think “apparatus and model” was a reminder to write about that. Please check, correct, and add missing information. Furthermore, what is the role of glycerol used during centrifugation? Is it a co-solvent? If yes, could solubilize even the NAP entrapped inside the nanoparticles, underestimating the results. Please clarify this part.
Lines 545-552: The paragraph when Authors described the Entrapment Efficiency methods is quite difficult to read because of the syntaxis is misleading. Furthermore, in the previous paragraph glycerol was used during centrifuge analysis while in this part it was not reported, why?
Line 572: Which are the formulations analysed by SEM named as “solid and liquid”? do they refer to formulations “in water dispersion” and “lyophilized”, respectively? written in this way is too generic.
Lines 581-582: The description of sieving method “sieving through #100” is too generic, it could be changed in “sieving through 100 mesh size sieves”. Furthermore, the phrase reported in line 582 between brackets seems out of context, please check, and eventually correct it.
Line 625: I think this sentence “spreading value is critical regarding gel formulation's therapeutic efficacy” needs to be supported by a reference.
Line 649: did the accelerated stability study performed on all the formulations or also on the optimized formulation? Please clarify it.
Reviewer 2 Report
In the present work, authors describe the preparation, characterization and biological evaluation of naproxen (NAP) nanoparticles made of chitosan and carrageenan and their incorporation into a gel to achieve a semi-solid formulation suitable for topical application. The work presents a large collection of scientific data, however, several methodological aspects were not explained, the rationale for some assays is not clear, some control groups are missing. Several animals were sacrificed for this work, even to achieve biological membranes for Franz diffusion cells. I suggest authors to check point-by-point the comments and suggestions to improve the presentation of the data. I recommend authors to resubmit the manuscript after the proper corrections/clarifications have been made.
Particular comments:
The title is not convincing. If they are novel structures why optimize? In this study, the histopathology is a tool used to confirm an in vivo finding and should not appear in the title. I suggest rephrasing the title.
Affiliation, Figure Captions: format improvement is needed
The general presentation of the results is quite confuse. Figure/Table captions are very much incomplete, as long as many abbreviations are presented in both figures and tables. Additionally, no mention is made to the statistical significance of the data. Table 2 is not comprehensive and does not replace individual information to be given in each graphic/table.
CS/CRG nanoparticle toxicity has been tested in vitro in a variety of cell lines. What was the rational to conduct an oral toxicity assay for a topical gel. Oral toxicity assay authorization number from the local ethical committee is missing. Regarding the oral toxicity, what received the “Control group” and what received the “Treatment group”, and how animals were administered? Methods are very incomplete.
In view of their potential topical application, the cytotoxicity of the prepared nanoparticles and/or the final gel should have been evaluated using HaCaT cells (an immortal human keratinocyte cell line). At least, have you checked any published data on this regard?
Why presenting results prior to the methods? Some abbreviations are presented in the Methods section but, for the reader, results appear first. Please, carefully check abbreviations and their order of appearance.
Moreover, if presenting results prior to the methods, the results headings should express the main findings of each assay, which is not the case.
Box-Behnken Design Analysis results should appear before the SEM, FTIR, pXRD data, as long as these last techniques were used for optimized formulation.
Currently, there are in vitro alternatives to in vivo Draize Skin Irritation test that could have been used. You have stated that the ethical committee approved the study. Please specify and insert the number of the authorization.
Separate numbers from units: eg. 2 mL instead of 2mL
Once abbreviated there is no need to repeat abbreviations (eg. Ca-940)
When writing the formula do not use dots before the formula number …… (x)
Homogenize time units: h, min, s (abbreviated) or hours, minutes, seconds (non-abbreviated)
Permeation: The amount of formulation applied ensured an infinite dose assay? Please, specify.
NP gel, np gel: use abbreviations only after full description, homogenize capital/non capital letters
In the final formulation (gel), what was the NAP loading? This would be very important to compare with a commercial NAP gel that contains 100 mg Naproxen per g of gel.
In vivo anti-inflammatory studies in rats:
- If animals are not anesthetized prior the topical application how do you ensure that animals will not clean or remove the formulation by pressing the paw on the flour.
- How do you inject formalin intradermally on non-anesthetised/unrestrained animals?
- What was the amount of formulation applied and what was the NAP gel control dose?
- Why 15 mg/kg? This dose was based on any reported study?
- How many formulation applications were made? The first dose was applied on hour before formalin injection. And the second? You wrote “continued till 5hs”. How many times? A scheme of the animal model would be very helpful.
- How (what technique was used) did you measure the paw edema? Please, specify the apparatus.
- Normal control is basically naïve animals needed for histological comparison. Do you need 6 animals in this group?
- Figure 9 and 10: you should mention that they are representative images of each group
- A commercial NAP topical formulation (usually 10% NAP) is missing as a positive control group.
Optimized formulation: sometimes you refer it as F0, sometimes as N0, other as No. Please, homogenize
Ln64: What are ADRs?
Ln70: What is PEC?
Ln86: What is the size of such particles, to be able to penetrate the skin epithelia?
Ln532: apparatus and model? Careful reading of the manuscript would have avoided nonsense words
Ln 532: 1200 pm or rpm?
Ln532: on a 10μL glycerol layer for 40 min. What does it mean?
Ln536: Carbopol 940 (0.1%) used as gelling agent. This is not necessary.
Ln613: …and housed for a sometimes… please, rephrase
Ln673: What are “opt. nanoparticles”?
Ln715: …Bahawalpur, Pakistan) moreover, fed … please, rephrase
Table 3: You should present Skin irritation “score” instead of “studies”
Table 3: What is N control?
Reviewer 3 Report
Sobia Noreen et al. reported in this paper the preparation and evaluation of naproxen loaded chitosan/carrageenan nanoparticles with improved permeation through topical route and anti-inflammatory activity. Сhitosan based nanoparticles attract significant attention as a drug carrier. They can accommodate drugs efficiently in their pore spaces, leading to high drug encapsulation efficiency and loading capacity, which is essential for prolonged drug release. In the present study the authors applied chitosan/carrageenan nanoparticles for topical delivery of naproxen, and showed that this formulation is effective in treating arthritis-associated inflammation. In addition, the acute oral toxicity study predicted that fabricated nanocarriers are useful for effective drug delivery through various routes. The presented manuscript is well written, the experiment was properly conducted, and the obtained results meet the requirements and readership for publication in Pharmaceutics.
Round 2
Reviewer 2 Report
The authors provided the revision details, however, changes cannot be easily identified in the revised manuscript. Some changes are written in red but they not represent all the modification made.
|
Reviewer 2 |
|
||
|
No. |
Comments |
Responses |
V2 comments |
|
1 |
The title is not convincing. If they are novel structures why optimize? In this study, the histopathology is a tool used to confirm an in vivo finding and should not appear in the title. I suggest rephrasing the title. |
Thanks for the correction, Now, the title of the manuscript has been changed from ‘Optimization of novel Naproxen loaded chitosan/ carrageenan nanocarrier based gel for topical delivery: ex vivo, histopathological and in vivo evaluation” to ‘Optimization of novel Naproxen loaded chitosan/ carrageenan nanocarrier based gel for topical delivery: ex vivo and in vivo evaluation’ Word optimize has been used because of box behnken design used for formulation development and one optimized formulation was obtained. |
OK. But the new title was not modified in the revised manuscript |
|
2 |
Affiliation, Figure Captions: format improvement is needed |
Affiliation and Figure Caption has been revised and corrected. |
OK |
|
3 |
The general presentation of the results is quite confuse. Figure/Table captions are very much incomplete, as long as many abbreviations are presented in both figures and tables. Additionally, no mention is made to the statistical significance of the data. Table 2 is not comprehensive and does not replace individual information to be given in each graphic/table. |
To enhance the clarity, abbreviations in figure captions and tables has been explained and presented in Table 1. The authors have added the [Table 2] to represent the ANOVA test values of dependent responses of Box-Behnken Design Analysis. As in this study Box-Behnken Design has been applied on formulations and statistical data of suggested variables has been explained in this table. |
I still do not know if data represent mean ± SD or mean ± SEM or median, etc. |
|
4 |
CS/CRG nanoparticle toxicity has been tested in vitro in a variety of cell lines. What was the rational to conduct an oral toxicity assay for a topical gel. Oral toxicity assay authorization number from the local ethical committee is missing.
Regarding the oral toxicity, what received the “Control group” and what received the “Treatment group”, and how animals were administered? Methods are very incomplete. |
Recently a study reported the oral toxicity assay for a topical gel (1)
Authorization number from the local ethical committee is (08-2020/PAEC) The Islamia University of Bahawalpur. Requested information is added. Control group receive no treatment and only Standard food and water was administered to Group-I while nanoparticle formulation was administered to Group II (Treatment group) in form of nanoparticle suspension via oral syringe. |
The oral acute toxicity assay of the mentioned publication was made for the proliposomes to be used for oral delivery of glibenclamide, not for the loaded liposomes to be used topically. |
|
5 |
In view of their potential topical application, the cytotoxicity of the prepared nanoparticles and/or the final gel should have been evaluated using HaCaT cells (an immortal human keratinocyte cell line). At least, have you checked any published data on this regard? |
Thanks for some valuable suggestions. In this project In vivo toxicity was tested. But mentioned cytotoxicity cannot be performed now as project is closed. But your valuable suggestion will be considered for our future projects. |
The suggestion is to refer the existing studies already published and not to perform an MTT assay. |
|
6 |
Why presenting results prior to the methods? Some abbreviations are presented in the Methods section but, for the reader, results appear first. Please, carefully check abbreviations and their order of appearance. Moreover, if presenting results prior to the methods, the results headings should express the main findings of each assay, which is not the case. |
According to the guidelines of Pharmaceuticals Journals it is recommended to present the Results and Discussion prior to methods sections Now the headings, order of appearance and all abbreviations has been clearly evaluated and revised in the manuscript. As authors applied box behnkhen design analysis therefore some characterization has been discussed under this design. |
I cannot find the modifications in the new version of the manuscript |
|
7 |
Box-Behnken Design Analysis results should appear before the SEM, FTIR, pXRD data, as long as these last techniques were used for optimized formulation |
Your suggestion is very valuable but a separate section is added for optimized formulation and if we add after SEM, FTIR, pXRD data after that the whole structure of article not seems suitable |
I do not agree. If the characterization is made on optimized formulations, the optimization must appear first. |
|
8 |
Currently, there are in vitro alternatives to in vivo Draize Skin Irritation test that could have been used. You have stated that the ethical committee approved the study. Please specify and insert the number of the authorization. |
Animal’s trial studies approved by the Pharmacy Animal Ethics Committee (08-2020/PAEC) The Islamia University of Bahawalpur |
OK |
|
9 |
Separate numbers from units: eg. 2 mL instead of 2mL |
Correction has been done in line number 495 |
There are other places to make modifications, line 495 was only an example. The manuscript was not completely checked on this regard. |
|
10 |
Once abbreviated there is no need to repeat abbreviations (eg. Ca-940)
When writing the formula do not use dots before the formula number …… (x) |
Corrections has been done as per recommendation of the reviewer. |
It is very difficult to identify the modifications made. Some dots were left on. |
|
11 |
Homogenize time units: h, min, s (abbreviated) or hours, minutes, seconds (non-abbreviated) |
Correction has been done and units are abbreviated throughout the manuscript. |
It is very difficult to identify the modifications made. |
|
12 |
Permeation: The amount of formulation applied ensured an infinite dose assay? Please, specify. |
The amount of formulation applied i.e. 10mg/cm2 of NAP NP gel ensured the infinite dosing assay (2) |
OK |
|
13 |
NP gel, np gel: use abbreviations only after full description, homogenize capital/non capital letters |
NP gel has been capitalize throughout the manuscript and abbreviated after full description. |
OK |
|
14 |
In the final formulation (gel), what was the NAP loading? This would be very important to compare with a commercial NAP gel that contains 100 mg Naproxen per g of gel. |
NAP nanoparticles containing gel equal to 5 mg NAP was used in franz cells for permeation studies. Final formulation gel was prepared by adding NAP containing nanoparticles (equal to 50 mg NAP) into carbopol 940 gel (100mg). Control gel was prepared by adding pure drug (NAP 50mg) into carbopol 940 gel (100mg). |
It means that you are testing a formulation with a NAP dose 5 times higher than the commercial form. How can your optimized formulation be an alternative to the commercial NAP gel? |
|
15 |
In vivo anti-inflammatory studies in rats:
If animals are not anesthetized prior the topical application how do you ensure that animals will not clean or remove the formulation by pressing the paw on the flour. |
We have mentioned in the manuscript that animal were anesthetized prior the topical application. We will further assure at the time of resubmission.
|
Where? I cannot find it. Is not version 2 a resubmission? |
|
16 |
How do you inject formalin intradermally on non-anesthetised/unrestrained animals? |
As mentioned earlier animals were anesthetized prior the topical application. |
|
|
17 |
What was the amount of formulation applied and what was the NAP gel control dose? |
NAP nanoparticles containing gel equal to 5mg naproxen in 10mg gel was applied as formulation. 5mg pure naproxen containing gel (10mg) was used as control gel. |
It means that you are testing a formulation with a NAP dose 5 times higher than the commercial form. How can your optimized frmulation be an alternative to the commercial NAP gel? |
|
18 |
Why 15 mg/kg? This dose was based on any reported study? |
As per recommendation of the reviewer, a reference supporting this sentence has been added on line no 667 (3) |
OK |
|
19 |
How many formulation applications were made? The first dose was applied on hour before formalin injection. And the second? You wrote “continued till 5hs”. How many times? A scheme of the animal model would be very helpful. |
Single treatment dose was applied. Correction has been made and the sentence rephrased as “The dose of treatment (NAP loaded NP gel) was applied topically 60 min before formalin injection”. (4) |
OK |
|
20 |
How (what technique was used) did you measure the paw edema? Please, specify the apparatus. |
The paw edema meter was used to measure the paw volume. We have already mentioned this in section 4. Materials and Methods line number 669. |
Paw edema meters do not exist. What exists is a plethysmometer! |
|
21 |
Normal control is basically naïve animals needed for histological comparison. Do you need 6 animals in this group? |
In order to minimize the chances of deviation we selected same no of animals (6) as for other groups. |
Because the inhibition is based on the volume of normal control animals (VC), you should say. Actually the formalin induced group is not a treatment group but a control (negative control group). Can you explain why FIE %inhibition has no deviation and has precisely the same value in all times evaluated? |
|
22 |
Figure 9 and 10: you should mention that they are representative images of each group |
These are the representative images of optimized formulation. In caption representative images of control and treatment group has been mentioned. |
I cannot see it. |
|
23 |
A commercial NAP topical formulation (usually 10% NAP) is missing as a positive control group. |
Author used Control gel in this study. Control gel was prepared by adding pure drug (NAP 50mg) into carbopol 940 gel (100mg). |
It does not replace the commercial formulation. |
|
24 |
Optimized formulation: sometimes you refer it as F0, sometimes as N0, other as No. Please, homogenize |
The correction has been done throughout the manuscript and now optimized formulation is referred as No. |
OK |
|
25 |
Ln64: What are ADRs? |
The word ADRs replaced with side effects in line 64 for the better understanding of readers. |
OK |
|
26 |
Ln70: What is PEC? |
PEC is the abbreviation of Polyelectrolyte complex. Correction has been done accordingly |
OK |
|
27 |
Ln86: What is the size of such particles, to be able to penetrate the skin epithelia? |
The optimum particle size for the permeation of nanoparticles through hair follicles was in the range of 400–700 nm (5).(6). For the absolute permeation of nanoparticles through the human skin, the nanoparticles' size should be less than 400 nm for drug delivery applications (7). This sentence was already added in results and discuusion section at line nmber 392. |
OK |
|
28 |
Ln532: apparatus and model? Careful reading of the manuscript would have avoided nonsense words |
Thanks for the correction. “Apparatus and model” number has been added. |
OK |
|
29 |
Ln 532: 1200 pm or rpm? |
1200 rpm. Correction has been done in the manuscript file. |
OK |
|
30 |
Ln532: on a 10μL glycerol layer for 40 min. What does it mean? |
Confusing words has been removed to clear the ambiguity |
OK |
|
31 |
Ln536: Carbopol 940 (0.1%) used as gelling agent. This is not necessary. |
Carbopol. 940 is commonly use as gelling agent in various nanoparticle based gel studies. |
I do know what Carbopol 940 is. The question is relative to the unnecessary short sentence. At least, if you want to keep it, it must be rephrased to “Carbopol 940 (0.1%) was used as the gelling agent.” |
|
32 |
Ln602: …and housed for a sometimes… please, rephrase |
Sentence has been rephrased and corrected. |
OK |
|
33 |
Ln673: What are “opt. nanoparticles”? |
opt. nanoparticles is referred to optimized nanoparticles formulation suggested by Box-Behnken Design. |
OK |
|
34 |
Ln715: …Bahawalpur, Pakistan) moreover, fed … please, rephrase |
Correction has been done accordingly. |
OK |
|
35 |
Table 3: You should present Skin irritation “score” instead of “studies” |
“Skin irritation studies” has been replaced with Skin irritation “score” instead of “studies” |
OK |
|
36 |
What is N control? |
This mistake in the table has been corrected and “N control” has been replaced with No. |
OK |
References:
- Khan S, Madni A, Rahim MA, Shah H, Jabar A, Khan MM, et al. Enhanced in vitro release and permeability of glibenclamide by proliposomes: Development, characterization and histopathological evaluation. Journal of Drug Delivery Science and Technology. 2021;63:102450.
- Lau WM, Ng KW. Finite and infinite dosing. Percutaneous Penetration Enhancers Drug Penetration Into/Through the Skin: Springer; 2017. p. 35-44.
- Cong H, Khaziakhmetova V, Zigashina L. Rat paw oedema modeling and NSAIDs: Timing of effects. International Journal of Risk & Safety in Medicine. 2015;27(s1):S76-S7.
- He Y, Majid K, Maqbool M, Hussain T, Yousaf AM, Khan IU, et al. Formulation and characterization of lornoxicam-loaded cellulosic-microsponge gel for possible applications in arthritis. Saudi Pharmaceutical Journal. 2020;28(8):994-1003.
- Lademann J, Meinke M, Sterry W, Patzelt A. Wie sicher sind Nanopartikel? Der Hautarzt. 2009;60(4):305-9.
- Ghasemiyeh P, Mohammadi-Samani S. Potential of Nanoparticles as Permeation Enhancers and Targeted Delivery Options for Skin: Advantages and Disadvantages. Drug Design, Development and Therapy. 2020;14:3271.
- Verma A, Jain A, Hurkat P, SK J. Transfollicular drug delivery: current perspectives. Research and Reports in Transdermal Drug Delivery. 2016;5:1—17.